# Research on an Active Hydrogen Maser Digital Circuit Control System Based on FPGA

**DOI:** 10.3390/s23229202

**Published:** 2023-11-15

**Authors:** Wangwang Hu, Tao Shuai, Yonghui Xie, Pengfei Chen, Yuxian Pei, Yang Zhao, Rui Wang

**Affiliations:** 1School of Communication and Information Engineering, Shanghai University, Shanghai 200444, China; 2Shanghai Astronomical Observatory, Chinese Academy of Sciences, Shanghai 200030, China

**Keywords:** active hydrogen maser, frequency stability, FPGA, digital circuit, temperature control

## Abstract

A hydrogen maser is a high-precision time measurement instrument with high frequency stability and low frequency drift, which is widely used in satellite navigation, ground time keeping, frequency measurement, and other fields. An active hydrogen maser (AHM) is better than the current space passive hydrogen maser (PHM) in orbit in terms of its frequency stability and drift rate, but it has the disadvantages of large volume and weight. To further reduce the volume and weight of the circuit, this paper demonstrates a digital circuit control system based on a field-programmable gate array (FPGA). It uses digital temperature control, digital detectors, digital down-conversion, digital phase-locked loops, and other digital methods for temperature control, cavity auto-tuning, and crystal phase locking, which improve the integration and flexibility of the circuit system. Meanwhile, a tuning method based on hydrogen flow is proposed, which effectively solves the problem of fluctuations in hydrogen maser resonance frequency with changes in the external environment. Our experimental results show that the designed digital circuit control system meets the requirements of an oven-controlled crystal oscillator (OCXO) loop and a cavity loop. Its frequency stability can reach $2.6\times 10^{-13}/1\;s$ and $1.4\times 10^{-15}/10,000\;s$, which is close to the stability index of ground active hydrogen maser. This scheme has certain practical engineering value, and can be used in the design of hydrogen masers for next-generation space navigation satellites, deep space exploration, and space stations.

## 1. Introduction

Space atomic clocks are the core components of modern navigation satellites. They determine the signal broadcasting performance of navigation satellites and the positioning precision of navigation systems [1]. The passive hydrogen maser is small in size and weight, and has excellent short-to-medium-term frequency stability and frequency drift. As a high-precision frequency source, it is widely used in satellite navigation systems such as Galileo and Beidou, providing strong technical support with which navigation satellites can provide precise positioning services and long-term ephemeris forecasts [2,3].

When excited by external signals, hydrogen atoms undergo a stimulated transition from the hyperfine energy level ($F=1,m_{F}=0$) to the hyperfine energy level ($F=0,m_{F}=0$), generating an inductive radiation signal of 1420.405751 MHz [4]. This signal is used to discriminate the output frequency of the OCXO. The crystal oscillator outputs a 10 MHz frequency reference signal with high stability, high accuracy, and a low drift rate by adjusting its control voltage [5].

Since there are certain gaps in the stability and drift rate of the passive hydrogen maser compared to the ground active hydrogen maser, it is difficult to support the high-precision timekeeping performance of satellites for durations on the order of tens of days [6]. Therefore, it is necessary to develop a space active hydrogen maser with higher performance indices. However, due to its large volume and weight, the ground active hydrogen maser is currently difficult to adapt to most satellite platforms represented by navigation satellites, so it needs lightweight improvements. At the Neuchatel Observatory in Switzerland, the mass of the entire system was significantly reduced to about 35 kg, and a frequency stability of $1.5\times 10^{-15}/10,000\;s$ was achieved [7,8]. SpectraTime developed a 40 kg lightweight active space hydrogen maser for the European Space Atomic Clock Ensemble in Space (ACES) mission, which was flown to the International Space Station (ISS). Its frequency stability reached $1.5\times 10^{-15}/10,000\;s$ [9]. The Beijing Institute of Radio Metrology and Measurement also developed a space active hydrogen maser with a mass of about 40 kg, which was launched with satellites in 2022 [10]. The lightweight and improved active hydrogen maser has the potential to be used in the next generation of navigation satellites and satellites related to positioning, navigation, and timing (PNT) technology. Combined with the satellite atomic clock group weighting, atomic clock steering [11], and joint timekeeping of space constellations, it can improve the spatial timekeeping performance of satellites and provide technical support for the long-term autonomous operation of satellites.

The circuit is an important component of the hydrogen maser, which performs the functions of temperature control, microwave cavity and crystal oscillator locking, and high-stability frequency signal output. Miniaturization, digitization, and integration of the circuit system represent the development trends in space product design. The temperature control circuit of the space passive hydrogen maser, the microwave cavity, and the crystal control circuit of the previous ground active hydrogen maser are analog control circuits, which are large, cumbersome to debug, and inconvenient to monitor and reconfigure in orbit. In response to this situation, we used a digital circuit based on FPGA for system control and explored a low-power, high-integration implementation to provide a reference design for the development of space active hydrogen masers.

## 2. Basic Principles of the Hydrogen Maser

### 2.1. Analysis of Frequency Stability of Hydrogen Maser

Frequency stability is a significant performance indicator for the hydrogen maser. In satellite navigation systems, the atomic clock’s frequency stability determines the fluctuations of satellite time relative to the predicted time during the ephemeris forecast period, and affects the precision and range of positioning for users.

The frequency stability of hydrogen masers in the time domain is determined by the Equation (1) [12,13]:(1)$$\sigma_{y}(\tau )=\frac{1}{Q_{a}\sqrt{\tau }}\sqrt{\frac{kT}{2P_{a}}}$$
where τ is the averaging time; k is the Boltzmann’s constant ($1.38\times 10^{-23}\;J/K$); $Q_{a}$ is the quality factor of the hydrogen atom spectral line; T is the absolute temperature of the hydrogen maser; and $P_{a}$ is the power delivered by the atoms to the cavity. This expression describes hydrogen maser frequency stability at an averaging time of τ>10 s. According to Equation (1), the frequency stability of a hydrogen maser depends on the averaging time, $\tau^{-1/2}$. Increasing the output power of the hydrogen maser can improve its frequency stability. However, its frequency stability will be affected by additional noise when the physical perturbation factors, such as the second-order Doppler effect, cavity-pulling effect, wall shift, static magnetic field [14], etc., begin to act. Reference [15] offers a quantitative analysis of the influence of these physical perturbations on frequency stability, as shown in Table 1 [15].

To achieve improved frequency stability, we adopted multi-layer temperature control and optimized thermal insulation design measures to ensure a constant temperature in the microwave cavity and to reduce the impact of the second-order Doppler effect. By improving the materials, coating process, and material aging, the influence of wall shift can be reduced. The stability of the static magnetic field can be ensured by optimizing the magnetic shield materials, processing techniques, and structural design, as well as utilizing multi-layer, high-performance magnetic shields [4]. The cavity-pulling effect is very important for achieving long-term frequency stability. Using the cavity auto-tuning method can maintain the stability of the microwave cavity’s central resonant frequency and reduce the influence of the cavity-pulling effect [5].

The microwave cavity is a vital component of a hydrogen maser and directly affects the performance index of whole machine. Its function is to store microwave radiation energy and provide the necessary conditions for interaction between the radiation field and atoms, as well as the feedback of stimulated radiation. The cavity-pulling effect causes the output frequency of a hydrogen maser to deviate from the atomic transition frequency when there is a difference between the resonant frequency of the microwave cavity and the atomic transition frequency [14]. The influence of this effect can be expressed by Equation (2) [14]:(2)$$\frac{\Delta v_{C}}{v}=\frac{Q_{C}}{Q_{a}}\times \frac{\left(v_{c}-v_{0}\right)}{v_{0}}$$
where $\Delta v_{C}$ is the change in the output frequency of the hydrogen maser; $\left(v_{c}-v_{0}\right)$ is the difference between the resonant frequency of the microwave cavity and the atomic transition frequency; $Q_{C}$ is the loaded quality factor of the microwave resonant cavity; and $Q_{a}$ is the spectral line quality factor of a hydrogen atom. Physical factors can easily affect the resonant frequency of the microwave resonant cavity. The dielectric constant of the storage bubble, the expansion of the microwave cavity material, and the thermal expansion of the cavity’s silver coating layer are all related to the temperature of the microwave cavity.

The second-order Doppler effect is independent of atomic motion direction, and the relative frequency shift of the hydrogen maser caused by the second-order Doppler effect can be expressed by Equation (3) [14]:(3)$$\frac{\Delta v_{D}}{v}=-1.4\times 10^{-13}T$$
where $\Delta v_{D}$ represents the change in the output frequency of the hydrogen maser due to the second-order Doppler effect. According to Equation (3), the relative frequency shift is directly proportional to the temperature, and temperature change has a significant influence on the frequency shift. The thermal conduction path between the storage bubble and microwave cavity is good. When the temperature approaches equilibrium, the temperature of the storage bubble is approximately equal to that of the microwave cavity. Therefore, it is helpful to keep the storage bubble at a constant temperature in order to achieve better frequency stability.

### 2.2. Functions of the Hydrogen Maser Circuit System

The hydrogen maser circuit is mainly composed of a main electronic circuit and an auxiliary electronic circuit. The main circuit part includes a 10 MHz crystal oscillator with a constant temperature, an isolation amplifier circuit, an up-conversion circuit, a down-conversion circuit, a digital servo, and a frequency synthesizer circuit, while the auxiliary circuit includes a constant temperature circuit, a high-voltage source, and a constant current source. The internal working principal block diagram is shown in Figure 1.

The circuit loop of the hydrogen maser mainly includes two parts: the crystal oscillator loop and the cavity loop [4]. The output signal from the physical system of the hydrogen maser passes through the down-conversion module; then, an intermediate frequency (IF) signal of 20.405 MHz is obtained. After digital-to-analog conversion, it enters the FPGA for data processing. In the crystal oscillator loop, the phase difference between the maser signal and the local signal is assessed to form an error signal. In the cavity loop, an error signal is formed by comparing the intensity difference between the left and right detection signals interacting with the microwave cavity. The FPGA generates two control signals through a control algorithm to adjust the voltages of the crystal oscillator and the microwave cavity varactor diode. After that, the frequency of the 10 MHz crystal oscillator is locked on the resonance transition spectrum of the hydrogen atom, and the resonant frequency of the microwave cavity is locked on the 10 MHz crystal oscillator.

To ensure the long-term stable operation of the hydrogen maser, the auxiliary electronic system is also required to ensure that the physical package remains in a stable state. This is so that the hydrogen maser can output a stable reference signal for a long time. The auxiliary electronics system consists mainly of the parts shown in Table 2.

### 2.3. Principle of Cavity Frequency Control

In order to suppress the influence of the cavity-pulling effect, the hydrogen maser has a varactor diode inside. By changing the voltage across the varactor diode, its capacitance can be adjusted, thereby adjusting the capacitance distribution parameters in the microwave cavity. Therefore, the resonant frequency of the microwave cavity can be adjusted to reduce the influence of the cavity frequency change on the output frequency of the hydrogen maser.

In contrast to the passive hydrogen maser, the active hydrogen maser has a higher Q value, typically around 40,000. It can independently maintain stimulated transitions in the microwave cavity without the need for external signals to excite the hydrogen atom transition, and the control is relatively simple. At present, the active hydrogen maser often uses external detection signal injection or the cavity frequency switching method for cavity auto-tuning (CAT), and uses a phase-locked system to lock the crystal oscillator.

The SOHM-4A active hydrogen maser of the Shanghai Astronomical Observatory (SHAO) uses the external detection signal injection method for cavity auto-tuning, and the mixed detection signal can be expressed by Equation (4) [16,17]:(4)$$f_{t}=f_{0}+f_{m}g(t)$$
(5)$$g(t)=\left\{\begin{array}{c} 1\begin{array}{cc} & 0\leq t \end{array}<\frac{nT}{2}\\ -1\begin{array}{cc} & \frac{nT}{2}\leq t<nT \end{array} \end{array}\right.$$
where $f_{t}$ is the frequency of the detection signal; $f_{0}$ is the frequency of the hydrogen atom transition; $f_{m}$ is the frequency modulation depth; g(t) is the periodic square wave; and T is the period of the square wave.

The principle of the external detection signal tuning method is shown in Figure 2 [17]. The detection signal is periodically injected into the microwave cavity. The amplitudes of the left and right detection signals are calculated through coherent demodulation, and then the amplitudes are accumulated by integration within the valid period of the detection signal. At the end of each detection cycle, the amplitude of the left and right detection signals is subtracted to obtain the amplitude difference. The amplitude difference $\Delta A_{c}$ is fed back as an error signal through the integrator to the varactor diode in the microwave cavity to adjust the cavity frequency. In this way, the long-term stability of the microwave cavity’s central resonant frequency can be ensured.

Both the CH1-75A active hydrogen maser in Russia and the MHM-2010 active hydrogen maser in the United States of America use the cavity frequency switching method for cavity auto-tuning [15,18].

The advantage of using the external detection signal method for cavity auto-tuning is that the power of the left and right detection signals can be higher than the power of the maser signal, and the signal-to-noise ratio (SNR) when demodulating the amplitude modulation signal is high. Its disadvantage is that the periodic changes in the left and right detection signals can cause disturbance to the maser signal, affect the locking of the crystal oscillator, and be unfavorable for the short-term frequency stability of the output frequency. The cavity frequency switching tuning method has almost no influence on the short-term stability of the output frequency [19]. Due to using only varactor diodes to control and switch the cavity frequency, there is no need for detection signal generation or up-conversion modules, which can simplify the circuit design and reduce electronic noise [20]. A disadvantage of this method is that varactor diodes are nonlinear, which affects the long-term stability of the output frequency.

## 3. Design of Digital Circuit Closed-Loop Control System

The digital circuit designed in this paper was based on the XC4VSX35 FPGA chip manufactured by the Xilinx company. Virtex-4 devices are user-programmable gate arrays with various configurable elements and embedded cores optimized for high-density and high-performance system designs [21,22]. The block diagram of the peripheral circuit with the FPGA as the core is shown in Figure 3: the power chips provide various power supplies to the FPGA; the peripheral clock circuit provides clock signals for FPGA; the refresh chip reloads and periodically refreshes the programs from the Flash chip to FPGA through the interface, facilitating in-orbit reconstruction; the JTAG circuit is used to download and debug the program; the analog-to-digital converter (ADC) chips sample and quantize analog quantization before transmitting the data to the FPGA; and, finally, the FPGA chip generates control quantities to drive the digital-to-analog converter (DAC) chips, digital switches, and heating modules.

We used four types of ADC chips and DAC chips. A high-speed 16-bit ADC chip was used to collect down-conversion signals, with a sampling rate of 90 MSPS. A low-speed 14-bit ADC chip was used to collect temperature differences, with a sampling rate of 1 MSPS. A high-speed 14-bit DAC chip was used to generate the DDS signal, with an operating frequency of 90 MHz. Two low-speed 16-bit DAC chips were used to generate the control voltage, with an operating frequency of 15 MHz.

Compared with analog circuits, digital circuit components are smaller in size and have lower power consumption, features which are conducive to integration. The prototype of the hydrogen maser’s new digital circuit is shown in Figure 4. The digital circuit reduces the power ripple by using low-dropout regulator (LDO) chips and selecting low-temperature-drift and low-noise electronic components, which effectively reduces the impact of digital noise on the frequency stability of the hydrogen maser.

### 3.1. Temperature Control

The hydrogen maser used a thermal control scheme, which combined the whole temperature control of the physical part with the local temperature control of the sensitive part. The whole temperature control of the physical part was achieved by placing heating wires on the outer shield to control the temperature. At the same time, the microwave cavity is a temperature-sensitive component, and was used to place heating wires. In this way, the heating wires on the outer shield and the cavity allowed for two-layer precision temperature control of the microwave cavity, which improved the temperature control precision of the microwave cavity. We controlled the temperatures of three parts of the physics package: the middle and bottom of the microwave cavity and the outer shield.

The temperature control circuit system in the hydrogen maser mainly consisted of a temperature data acquisition module, a controller module, and a driving heating module. The block diagram of the temperature control system is shown in Figure 5. The temperature acquisition module included a Wheatstone bridge circuit, a multiplexer circuit, and an amplifier circuit. An eight-select-one channel selector selected the signal path and output it to the amplifier. A 14-bit ADC chip sampled the signal; then, the sampled data entered the FPGA. At the target temperature, the 1 Ω change in the thermistor caused the output voltage to change by about 0.0016 V, which reflects a temperature change of 0.0014 °C. The resolution of the temperature control ADC was 0.0003 V, and the 1 Ω change of the thermistor could be completely identified. The temperature sampling time at each part was 2 s, and the temperature control time constant was on the order of hundreds of seconds. The volume and mass of the hydrogen maser were relatively large, and the temperature changed slowly. The influence of the differential term was very small, and only PI control was used in the experiment. The FPGA chip was the core of the controller module; it processed the sampled signal data and generated pulse width modulation (PWM) control signals.

The temperature control system based on PWM had a simple circuit, convenient debugging, and strong anti-interference ability, but one of its disadvantages was that it easily generated high ripples that affected the stability of the control voltage. Therefore, it was necessary to use a filter circuit to convert the PWM signal into an equivalent DC voltage control signal. According to theoretical calculations and simulation analysis, the attenuation of PWM signal ripples by the second-order RC filter was more than twice that of the first-order filter. The second-order RC filter had smaller ripples and a better high-frequency filtering effect. Therefore, in this paper, a second-order RC filter was used to filter the output PWM control signal. The frequency of the PWM control signal was 3.7 kHz, and the cutoff frequency of the second-order RC filter was 160 Hz. After passing through the second-order RC filter, the signal ripple was reduced from 3.3 V to 37.5 mV, and the attenuation was −38.7 dB. After RC filtering, the maximum DC voltage was 3.3 V, and the resistance of the heating wire was about 100 Ω. However, the hydrogen maser had a large heat capacity, and the heating power generated by 3.3 V was small. The drive heating module used a DC-DC converter to convert the input voltage to produce an appropriate heating voltage and heating power. After using the DC-DC converter, the maximum heating voltage was able to reach 26 V.

### 3.2. Cavity Frequency Control

Due to the influence of the cavity pulling effect, the change in the central resonant frequency of the microwave cavity led to an output frequency change, resulting in the deterioration of the frequency stability. The typical value was $Q_{c}=40,000$, $Q_{\iota }=1.4\times 10^{9}$. If the frequency stability was required to be less than $1\times 10^{-15}/d$, the change in the central resonant frequency of the microwave cavity would have needed to be less than 0.05 Hz/d. However, the resonant frequency changes due to temperature fluctuations in the microwave resonant cavity and changes in the cavity structure were much greater than 0.05 Hz/d. Therefore, it was necessary to control the central resonant frequency of the microwave cavity to maintain the stability of the resonant frequency.

The schematic diagram of the closed-loop control system of the active hydrogen maser, which used the external detection signal tuning method, is shown in Figure 6. The digital circuit output detection signals of (1,420,405.751 ± 25) kHz through the upconverter. The output signals from the hydrogen maser were mixed and amplified by the downconverter. After ADC sampling, the intermediate frequency signals of 20.405 MHz entered the FPGA for digital filtering and synchronous detection processing. Then, the amplitudes of the left and right detection signals that passed through the cavity were detected, and the amplitude difference could be calculated. After passing through the servo module, the control signal was output through DAC to the varactor diode in the physical part of the hydrogen maser, thereby achieving the locking of the central resonant frequency of the microwave cavity.

When using the external signal detection method for cavity auto-tuning, it was necessary to obtain the intermediate frequency signal of 20.405 MHz, which was generated by direct digital synthesis (DDS) technology [23,24]. The DDS technology was based on the sampling theory. The analog standard sinusoidal signal was sampled by a working clock, and the amplitude value corresponding to the waveform phase was recorded to form a phase amplitude relationship table, which was stored in the waveform memory. During this time, the phase increased, and the amplitude changed periodically [24,25]. The frequency resolution of the DDS output signal was $3.2\times 10^{-7}\mathrm{Hz}$. After the sampled data from the ADC chip entering the FPGA, coherent demodulation was used in the FPGA to demodulate the amplitudes of the left and right signals, respectively. The local carrier referred to sine waves of (20,405.751 ± 25) kHz generated by DDS technology, which had the same frequencies as the left and right detection signals. The speed of microwave cavity control tuning was related to the period of the detection signal. An amplitude difference was generated at the end of each detection period, and the corresponding control amount was generated based on this amplitude difference. The control period which we used was 0.5 s. The system block diagram of the coherent demodulation method is shown in Figure 7. Since the frequency of the detection signal was known, the local in-phase and quadrature signals of the same frequency were generated by the DDS module of the FPGA. The high-frequency components in the signal after multiplication were filtered out using a digital low-pass filter. By adjusting the local phase using a digital phase-locked loop, locking was achieved, and the amplitude information of the AM signal was demodulated.

### 3.3. Phase Lock Control

The phase-locked loop is a closed-loop phase-tracking system with good narrowband tracking characteristics that can extract signals in low-SNR environments. In the crystal oscillator locked loop, the frequency and phase of the crystal oscillator’s output signal were synchronized with the frequency and phase of the hydrogen maser’s output signal by using the digital phase-locked loop technique. The frequency of the maser signal after down-conversion was 20.405751 MHz. After amplifying and sampling the signal, the sampled data were transferred to the FPGA. The FPGA used DDS technology to generate the 20.405751 MHz carrier signal. Since the frequency of the crystal oscillator was not the standard 10 MHz, the frequency phase difference $x\left(n\right)$ could be obtained by filtering the results after multiplying the carrier signal and the sampling signal.

Equations (6) and (7) [26] are often used to calculate the parameters of digital phase-locked loops in engineering practice:(6)$$c_{1}=\frac{4(w_{n}T{)}^{2}+8\xi w_{n}T}{4+(w_{n}T{)}^{2}+4\xi w_{n}T}\frac{1}{K_{z}}\approx \frac{2\xi w_{n}T}{K_{z}}$$
(7)$$c_{2}=\frac{4(w_{n}T{)}^{2}}{4+(w_{n}T{)}^{2}+4\xi w_{n}T}\frac{1}{K_{z}}\approx \frac{(w_{n}T{)}^{2}}{K_{z}}$$
where $K_{z}$ is the gain of the digital phase-locked loop; $w_{n}$ is the resonant frequency of the system; ξ is the damping coefficient, often taken as 0.707; and T is the sampling period [26]. The parameters $c_{1}$ and $c_{2}$ were calculated according to the above equations after the loop bandwidth was determined. The parameters were then adjusted in the closed-loop system based on these references, and the most appropriate parameters were selected according to the actual convergence situation. Then, the formula $S\left(n\right)=c_{1}\times x\left(n\right)+c_{2}\sum_{i=0}^{n-1}x\left(i\right)$ was used to generate the control quantity, thereby controlling the crystal oscillator and locking the frequency phase of the crystal oscillator to the atomic transition frequency phase.

The loop parameters of analog phase-locked receiver system were complicated to debug, and the adjusted parameters were fixed. Using the digital phase-locked loop, it was not only convenient to adjust the parameters, but we were also able to improve the performance of the system by dynamically adjusting the parameters $c_{1}$ and $c_{2}$. Usually, the capture band of the system is widened by increasing the resonant frequency of the system in the initial locking stage. The loop parameters calculated according to the equations were large, which made it easier for the system to enter the locking state quickly. After the system was locked for a period of time, the ability of the system to suppress noise was increased by reducing the noise bandwidth of the system. The calculated loop parameters were small, and the system was still able to maintain locking.

The locking loop of the crystal oscillator was analyzed and debugged by ChipScope Pro, an internal logic analyzer of Xilinx Company in San Jose, California, USA. The error curve of the crystal oscillator loop is shown in Figure 8, and the crystal oscillator was locked within 1 s. The frequency stability of the hydrogen maser at 1 s was determined by the crystal oscillator, thus requiring the crystal oscillator to be locked within seconds. As the error between the crystal oscillator loop and the cavity loop gradually decreased to 0, the hydrogen maser was in a locked state, and the circuit output a 10 MHz frequency signal with high stability and accuracy.

### 3.4. Nickel Purifier Tuning Control

Hydrogen storage provides high-purity industrial hydrogen, but it still contains impurities and, therefore, needs to be filtered and purified by a nickel tube purifier. In a vacuum environment, when the nickel tube is about 500 °C, hydrogen molecules can diffuse and penetrate through these metals from high pressure to low pressure while other molecules cannot. Thus, it serves as a hydrogen purifier.

In the process of using a nickel tube purifier, it is generally heated by applying a constant current. When its temperature is within an appropriate range, the purification and permeation of hydrogen gas can be achieved, allowing it to enter the ionization bubble and be ionized by the ionization source. The hydrogen purification efficiency of the nickel tube purifier varies at different working temperatures. And the temperature of the nickel tube purifier can be changed to control the hydrogen flow by adjusting its working current.

The output frequency of a hydrogen maser fluctuates due to changes in the external environment. And the output frequency is strongly related to the maser’s power. Therefore, it can be determined whether the tuning of the microwave resonant cavity is ideal by observing the fluctuation of the output frequency at different maser power levels. The most effective way to change the maser’s power is to adjust the hydrogen flow intensity by adjusting the operating current of the nickel tube purifier. At different resonant frequencies of the microwave cavity, the variation in the output frequency of the hydrogen maser varies with the same variation in the hydrogen flow rate. Therefore, an ideal turning point can be found to minimize the influence of hydrogen flow rate changes in the output frequency of a hydrogen maser.

### 3.5. Software Control System

According to the functional requirements of the closed-loop control system, the design of the software system was completed on the FPGA chip. The software can be divided into functional modules such as signal demodulation, crystal oscillator control, cavity frequency control, state machine, timing generation, DDS control, temperature detection, temperature control, PWM control voltage output, etc. Part of the software’s architecture is shown in Figure 9:

## 4. Experiment and Analysis

The hydrogen maser is a high-precision instrument. Some changes in the telemetry data during the experiment were very small, requiring a data acquisition instrument with high precision. We used a high-precision data acquisition instrument to collect and store the data.

The VCH-314 frequency comparator was used to test the frequency stability. The VCH-314 frequency comparator is intended for precise measurements of phase and frequency instability. It contains two identical measuring channels. Using the cross-correlation technique allowed us to obtain a super-low measurement error and to calculate the frequency instability of each signal separately. In this paper, the 10 MHz output signal from the Russian ground active hydrogen maser VCH-1003 M was used as the reference signal, and the locked 10 MHz standard output signal from the new lightweight active hydrogen maser was used as the measured signal, as shown in Figure 10.

### 4.1. Experimental Analysis of Constant Temperature Control

In this paper, the designed digital circuit was used to control the temperature of the new active hydrogen maser. Thermistors were used to collect the temperatures of the laboratory environment, microwave cavity, and outer shield in real time. The resistance values and three heating voltages were collected using a high-precision data acquisition instrument with a sampling interval of 4 s. According to the conversion formula between the thermistor value and the temperature, the results were as follows:

Figure 11a shows the temperature variation in the microwave cavity of the new, lightweight active hydrogen maser. The acquisition time was 116 h, the highest temperature was 49.7880 °C, the lowest temperature was 49.7804 °C, and the temperature variation was 0.0133 °C. Figure 11b shows the temperature variation in the outer shield of the new, lightweight active hydrogen maser, where the highest temperature was 39.1039 °C, the lowest temperature was 38.8834 °C, and the temperature variation was 0.2205 °C. Figure 11c shows the temperature variation in the laboratory environment, where the highest temperature was 26.2684 °C, the lowest temperature was 24.2689 °C, and the temperature variation was 2.0 °C.

As can be seen from Figure 11c, the temperature variation in the first 70 h had an obvious periodicity, which is related to the ambient temperature. The daily temperature of the laboratory changed by about 0.5 °C, which caused a change in the outer shield temperature of about 0.1 °C and a change in the microwave cavity of about 0.01 °C. After analyzing the stable temperature data, we found that there was more heat exchange between the outer shield and the external environment, as shown in Table 3, and the temperature change was relatively large due to the influence of the environment. The temperature control through the outer shield effectively reduced the influence of external temperature fluctuation on the microwave cavity, and better maintained the stability of the internal cavity’s temperature. The hydrogen maser was installed on a temperature-controlled satellite platform. Its temperature changed very little and its periodic fluctuations were not obvious, which is conducive to the hydrogen maser achieving good frequency stability.

### 4.2. Experimental Analysis of Nickel Purifier Tuning

The designed digital circuit was used to control the new lightweight active hydrogen maser, as shown in Figure 12. At the same time, the high-precision digital acquisition instrument was used to collect telemetry data such as maser power, light intensity, CAT voltage, and OCXO voltage.

We used VCH-314 to assess the frequency of the 10 MHz output signal from the new lightweight active hydrogen maser, and the sampling interval was 1 s. The collected data were processed in Stable32 to compare the frequency change before and after the nickel purifier tuning. Prior to nickel purifier tuning, the nickel purifier current was adjusted from 2.9 A to 2.2 A, and then subsequently from 2.2 A to 2.9 A, which caused a frequency change of about $6.7\times 10^{-12}$, as shown in Figure 13a. After tuning, the same change in the nickel purifier current resulted in a frequency change of about $9.5\times 10^{-14}$, as shown in Figure 13b. It could be seen that the nickel purifier tuning greatly reduced the influence of the hydrogen maser’s output frequency with the variation in hydrogen flow.

### 4.3. Experimental Analysis of Frequency Stability

Figure 14 shows the comparison results of a longer frequency stability test. The sampling interval of the VCH-314 was 1 s, and the comparison time was more than 420,000 s. In total, 42 million seconds fully achieved confident frequency stability at 10,000 s. The temperature change in the microwave cavity during this period is shown in Figure 11a. The short-to-medium-term frequency stability of the closed-loop test was good, the frequency stability reached $2.6\times 10^{-13}/1\;s$ and $1.4\times 10^{-15}/10,000\;s$. However, the frequency stability curve showed an upward trend starting from 10,000 s due to frequency drift, as shown in Figure 14a. The frequency stability curve after deducting the frequency drift using Hadamard variance is shown in Figure 14b.

The weight of the ground active hydrogen maser SOHM-4A was approximately 300 kg, and the power consumption was less than 250 W. The weight and power consumption of the new active hydrogen maser were significantly reduced using a dielectric microwave cavity with a higher quality factor for the physical package by designing modular digital systems for the circuit part. The weight was about 23 kg, and the steady state power consumption was about 60 W, which are not much different from the 23 kg space passive hydrogen maser. The typical indices of the ground active hydrogen maser SOHM-4A and the space passive hydrogen maser of SHAO are shown in Table 4. The comparison shows that the test performance indices of the new active hydrogen maser were better than the 23 kg space level passive hydrogen maser, and slightly lower than the ground active hydrogen maser SOHM-4A. The long-term stability of a lightweight AHM controlled by an analog circuit is slightly worse than that of a digital circuit. Although the noise of the analog circuit is relatively small, digitization of the hydrogen maser circuit may reduce the delay fluctuations caused by analog filtering and other parts, which has certain advantages in terms of improving long-term stability.

## 5. Conclusions

In this paper, a digital circuit control method based on FPGA was proposed, and a set of system software was designed to achieve closed-loop control of the hydrogen maser. We were able to maintain the stability of the microwave cavity temperature and resonant frequency. The digital circuit greatly improved the integration and flexibility of the circuit system through modularization and digitization, and further reduced the volume and weight of the circuit. The experimental results show that the temperature change in the microwave cavity of hydrogen maser was less than 0.014 °C within 116 h, and the external ambient temperature change did not exceed 2 °C. The frequency stability of the new hydrogen maser reached $2.6\times 10^{-13}/1\;s$ and $1.4\times 10^{-15}/10,000\;s$ when external detection signals were used for cavity auto-tuning. This scheme verified the feasibility of the digital circuit closed-loop control system, and the performance index was good. The digital circuit will be tested for long-term reliability and stability in order to better suit this new lightweight active hydrogen maser.

## Figures and Tables

**Figure 1 sensors-23-09202-f001:**
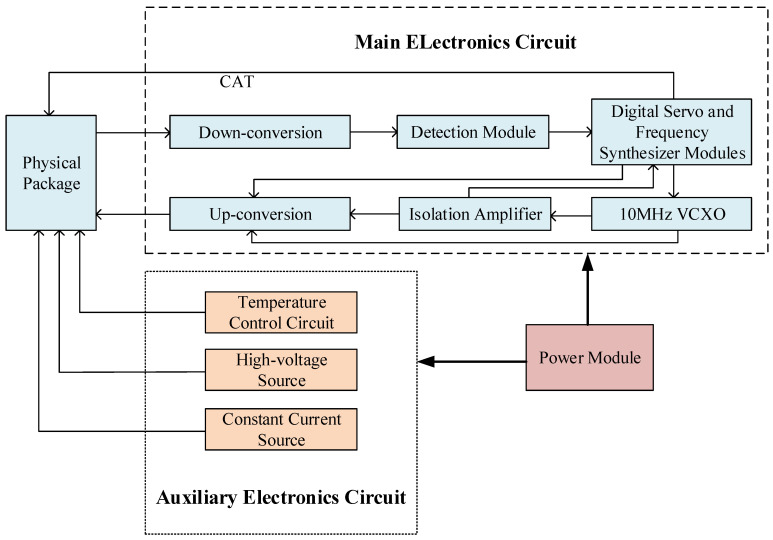
The operation principle of the circuit package.

**Figure 2 sensors-23-09202-f002:**
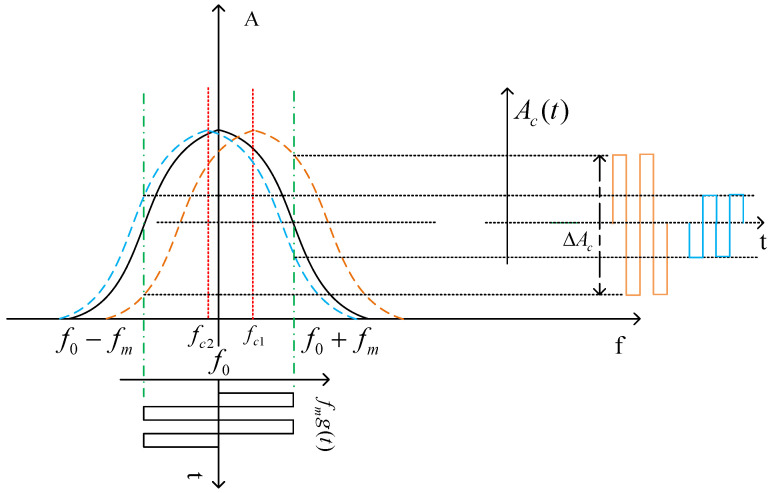
Schematic diagram of the external detection signal method.

**Figure 3 sensors-23-09202-f003:**
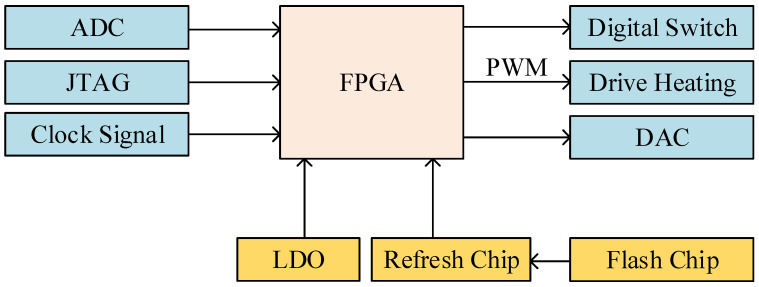
FPGA and external circuit interfaces.

**Figure 4 sensors-23-09202-f004:**
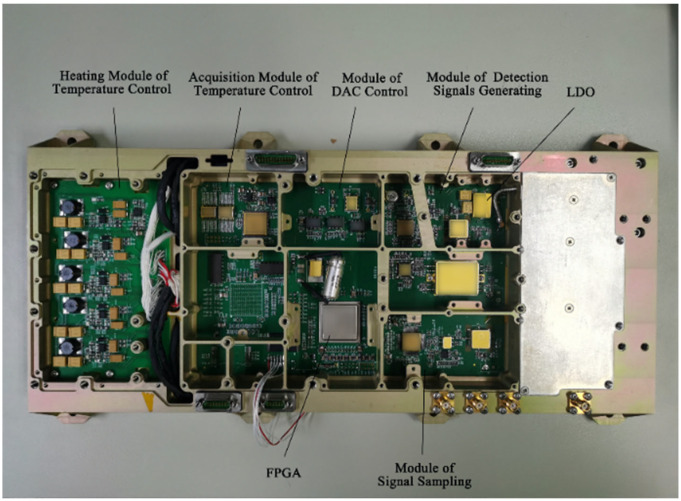
The prototype of the digital circuit.

**Figure 5 sensors-23-09202-f005:**
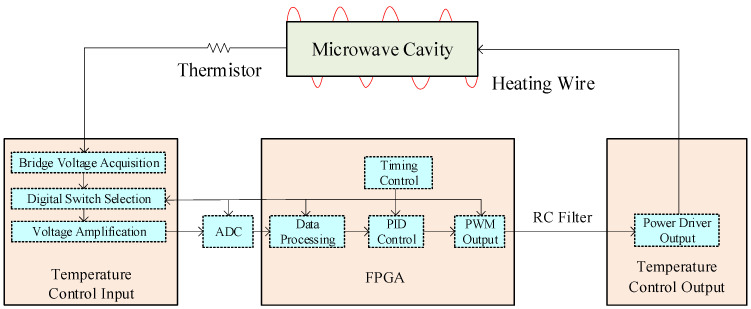
Structure of the temperature control system.

**Figure 6 sensors-23-09202-f006:**
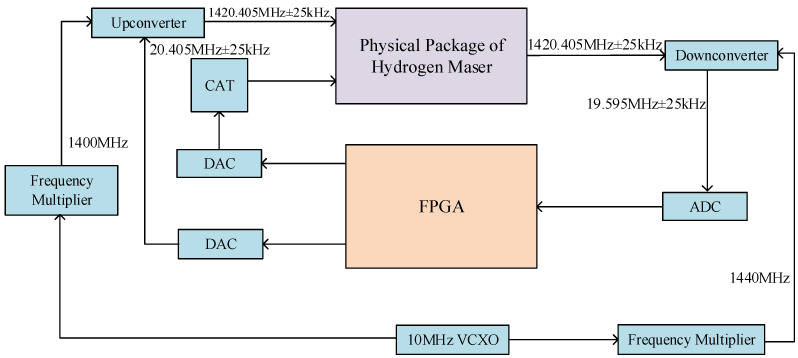
Schematic diagram of a cavity tuning system based on the external detection signal method.

**Figure 7 sensors-23-09202-f007:**
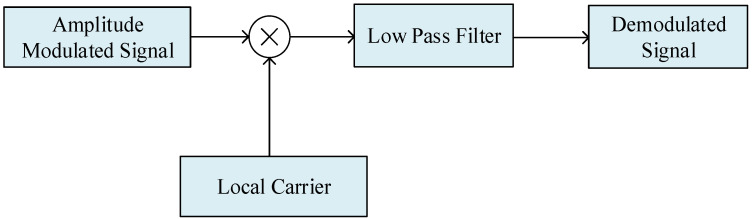
Block diagram of a coherent demodulation system.

**Figure 8 sensors-23-09202-f008:**
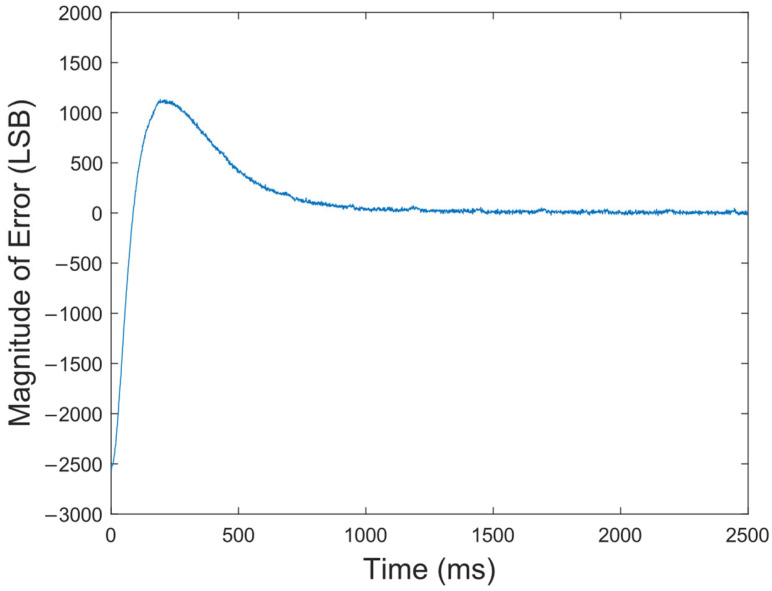
Error curve of the crystal oscillator loop.

**Figure 9 sensors-23-09202-f009:**
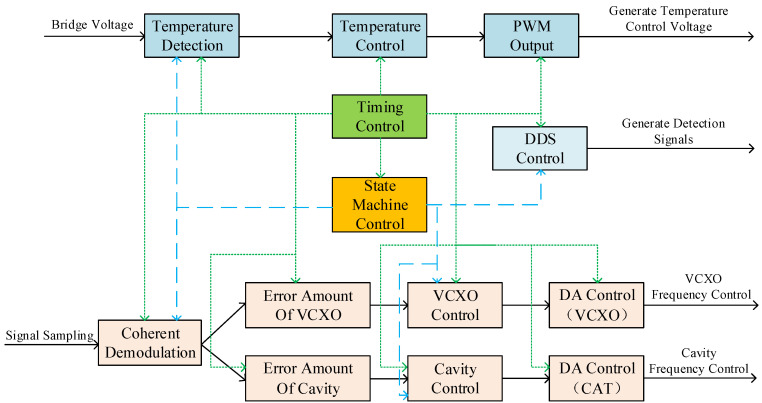
Software architecture of the closed-loop control system.

**Figure 10 sensors-23-09202-f010:**
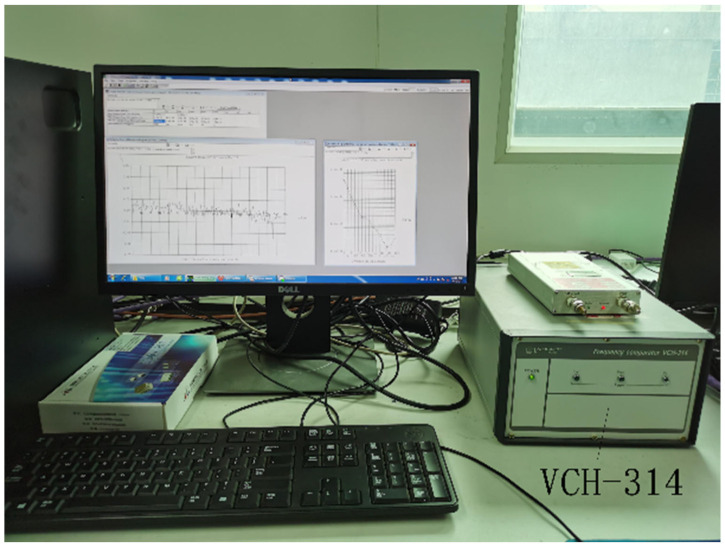
VCH-314 frequency stability test.

**Figure 11 sensors-23-09202-f011:**
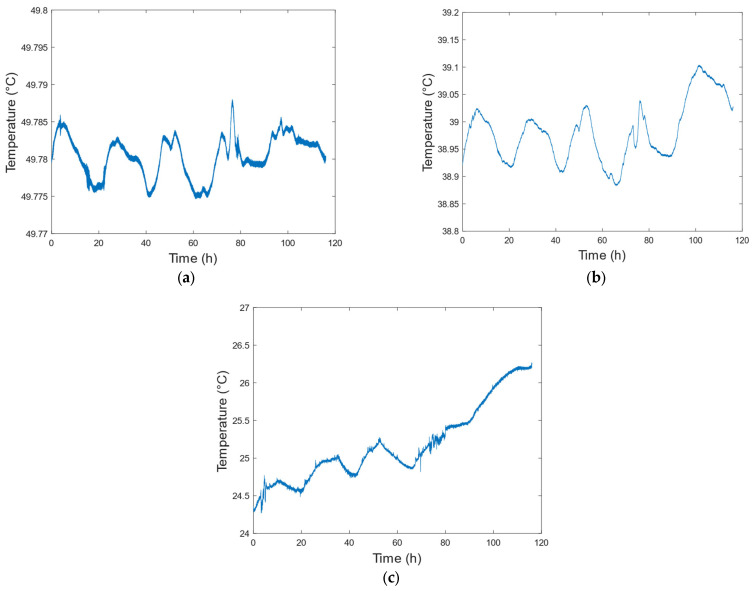
Temperature variation curves of (**a**) cavity; (**b**) outer shield; and (**c**) laboratory.

**Figure 12 sensors-23-09202-f012:**
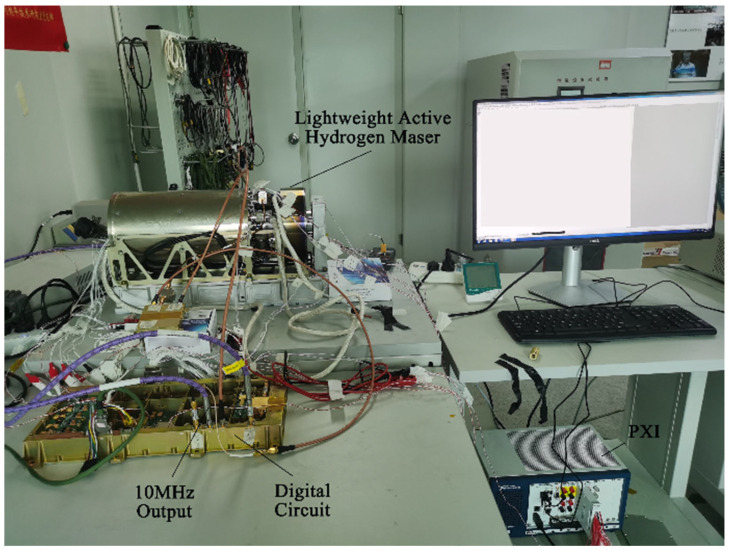
System closed-loop control test.

**Figure 13 sensors-23-09202-f013:**
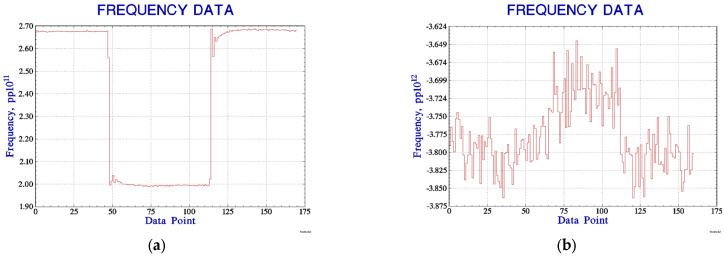
(**a**) Frequency variation before tuning; (**b**) frequency variation after tuning.

**Figure 14 sensors-23-09202-f014:**
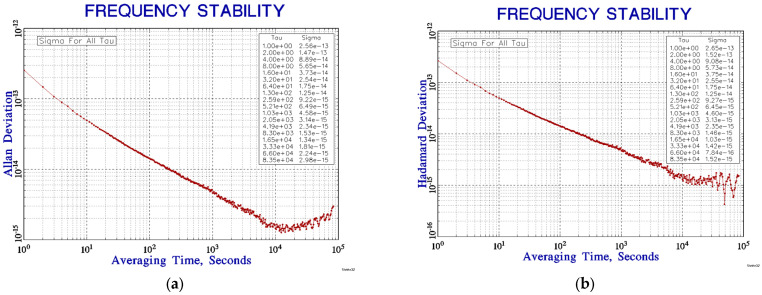
(**a**) Allan variance curve; (**b**) Hadamard variance curve.

**Table 1 sensors-23-09202-t001:** Hydrogen maser stability perturbation factors.

Perturbation Factors	Equation	Fractional Frequency Offset	Stability Floor
Second order Doppler’s effect	$\frac{\Delta v_{D}}{v}=\frac{-3\mathrm{kT}}{2{\mathrm{mc}}^{2}}$	$-4.31\times 10^{-11}$	$10^{-16}$
Spin-exchange shift	$\frac{\Delta v_{D}}{v}=\alpha n_{a}$	$2\times 10^{-13}$	$10^{-15}\sim 10^{-16}$
Magnetic field dependence	$\frac{\Delta v_{M}}{v}=\frac{2.773\times 10^{11}\times H_{0}{}^{2}}{v_{0}}$	$2\times 10^{-13}$	$10^{-15}\sim 10^{-16}$
Wall shift	$\frac{\Delta v_{W}}{v}=\frac{K}{D}$	$-1\times 10^{-11}$	$10^{-15}\sim 10^{-17}$
Cavity pulling	$\frac{\Delta v_{C}}{v}=\frac{Q_{C}}{Q_{a}}\times \frac{v_{c}-v_{0}}{v_{0}}$	≈0	$10^{-14}\sim 10^{-16}$

**Table 2 sensors-23-09202-t002:** Components and functions of the auxiliary electronics system.

Name	Function
Temperature control circuit	Ensure temperature stabilization of the temperature-sensitive components of the physical and electronic systems
Constant current source	Provide a stable current for the nickel purifier
High voltage source	Provide the high voltage required by titanium ion pumps to maintain high vacuum
Power source	Provide various power supplies for the whole machine

**Table 3 sensors-23-09202-t003:** Analysis of temperature control data.

Position	Average Temperature (°C)	Mean Square of Temperature (°C)	Temperature Variation (°C)
Cavity	49.7804	0.0027	0.0133
Outer shielding	38.9826	0.0542	0.2205
Laboratory	25.1764	0.5031	1.9995

**Table 4 sensors-23-09202-t004:** The frequency stability index of the hydrogen maser.

Performance	23 kg Space PHM	Lightweight AHM (Digital Circuit)	Lightweight AHM (Analog Circuit)	SOHM-4A
Allan variance				
1 s	$9.3\times 10^{-13}$	$2.56\times 10^{-13}$	$2.46\times 10^{-13}$	$2\times 10^{-13}$
10 s	$3.1\times 10^{-13}$	$4.93\times 10^{-14}$	$4.74\times 10^{-14}$	$4\times 10^{-14}$
100 s	$9.9\times 10^{-14}$	$1.41\times 10^{-14}$	$1.40\times 10^{-14}$	$9\times 10^{-15}$
1000 s	$2.9\times 10^{-14}$	$4.88\times 10^{-15}$	$6.62\times 10^{-15}$	$3\times 10^{-15}$
10,000 s	$1.0\times 10^{-14}$	$1.40\times 10^{-15}$	$2.13\times 10^{-15}$	$1\times 10^{-15}$

## Data Availability

The datasets generated and analyzed during the current study are available from the corresponding author upon reasonable request.

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
