# Peer review of "Research on an Active Hydrogen Maser Digital Circuit Control System Based on FPGA"

_sensors, 2023, doi:10.3390/s23229202_

Round 1

Reviewer 1 Report

Comments and Suggestions for Authors

The scientific value of the manuscript is not high.

The presented engineering work is more on a prototyping level and it is not ready yet for usage in space navigation satellites.

Comments on the Quality of English Language

Minor typos, not critical.

Author Response

Thank you very much for your attention and valuable comments on our paper. This manuscript has been revised carefully according to your comments. And we hope the explanations can fully addressed all of your concerns.

Reviewer 2 Report

Comments and Suggestions for Authors

The manuscript focuses on the development of a digital circuit control system for an Active Hydrogen Maser (AHM) based on FPGA. The paper provides detailed information on the design and implementation of the system, including the hardware and software components, as well as the tuning method based on hydrogen flow proposed in the study. The paper also discusses the advantages of using an AHM over a Passive Hydrogen Maser (PHM) in orbit, and explains how the digital circuit control system improves the integration and flexibility of the circuit system.

Although the manuscript covers all the necessary aspects, it lacks some detail in the description of the hardware, its performance, and its functionalities. It includes several block diagrams, which, however, are not sufficient to clearly represent the implemented functionalities, as they are often either too general or specific to a particular function. Consequently, the overall picture is not 100% conveyed. For instance, regarding the phase locking of the local oscillator, it does not indicate the signal conversion chain of the atomic signal, how it is acquired by the FPGA, and how the intermediate frequency phase is detected within the FPGA by the digital phase detector. While general usage formulas (9) and (10) are provided, it does not explain how the parameters c1 and c2 are used within the FPGA to calculate the frequency correction for the local oscillator. This way of describing the FPGA-based system weakens the manuscript and makes it less effective for potential readers. In my opinion, it would be sufficient to eliminate, reduce, or refer to concepts that are already widely known by the reader to allocate more space to the actual innovations presented in the manuscript. To assist the authors in this revision and improve the manuscript, I will provide some guidance on specific points. The authors can then adapt the same approach to the rest of the manuscript.

-        Sec2.3: principle of cavity frequency control. Here, the two commonly used methodologies are described. The authors should present only the technique actually implemented in this case and cite only the one that they do not actually use. On the other hand, they could provide more details on how the amplitude difference is processed to calculate the correction to the varactor (line 207).

-        Sec. 3: Please provide additional information: the FPGA size, the number of ADCs and DACs, and, if possible, the names of the chips used. At the very least, specify the number of bits, equivalent bits, and sample rate...

-        Sec. 3.1: In this case as well, more details are needed to reinforce the section: which parts of the physics package are controlled and the number of channels that reach the FPGA; the temperature accuracy that the sensors and their interface circuits are capable of supporting; controller time constants and PWM characteristics: operating frequency and RC filter cutoff.

-        Sec. 3.2: The equation (8) is well-known and can be omitted, as well as table 3. On the other hand, a more detailed description of how the information from the ADC is processed to obtain the received signal amplitude information can be provided. Additionally, it would be useful to better clarify what is meant by 'local carrier': its frequency and whether it is a square-wave or sine wave. Finally, specifying the speed (or bandwidth) at which the controller retunes the microwave cavity would be helpful.

-        Sec. 3.4: Figure 10, which discusses the software architecture, seems to not belong in this section, which is focused on the Nickel Purifier Tuning Control. It should be framed in a separate section or framed in the preceding sections that cover the specific functionalities. In this process, the authors should consider the partial overlap between figure 10 and the preceding ones, especially with figure 7.

Comments on the Quality of English Language

---

Author Response

(The authors gave the same response as above.)

Reviewer 3 Report

Comments and Suggestions for Authors

Dear Authors,

The article shows an electronic circuits based on FPGA that are used to control active hydrogen maser that could be used for space mission. But presented methods are not so novel but I understand that proposed system is essential step in space mission project. 

I found points that the article can be much better:

1) All results are not connected strongly quantitatively with the quality of the stabilization of the frequency. Please include this information in section 4. What parameter has the largest and smallest impact for the stability of the frequency? 

2) Table 5 presents a comparison between various hydrogen masers. There is no information about stabilization methods of the 23kg Space PHM and SOHM-4A. Is it used the same FPGA circuits or not? It is worth also include self-comparison of the Lightweight AHM with analog and digital stabilization. In the article there is no information why digital stabilization is better than analog. 

3) Please add information about conversion resolution of FPGA.

4) The PID control loops could be driven in various ways in FPGAs. Please show a simple mathematical model that you use in your circuits.

5) There is a mistake in the table 4 in average temperature of the laboratory. For sure it is not 24.2689 degC.

6) I do not understand the label axis in the figure 14. Usually the unit of frequency is Hertz.

Comments on the Quality of English Language

Quality of English requires a moderate editing in terms of grammar and vocabulary.  The style of the article should be improved. It cannot look like a report from student physical laboratory.  

Author Response

(The authors gave the same response as above.)

Reviewer 4 Report

Comments and Suggestions for Authors

      The paper studies the key technologies for implementing digital circuits of hydrogen maser,including temperature control circuit, cavity frequency locking circuit, phase locking control, etc. Based on the new dielectric microwave cavity physics system, the physical and circuit closed-loop control of a 23kg active hydrogen maser prototype has been achieved, and a short-term stability within 10000 seconds has been preliminarily obtained.

    This digital circuit implementation scheme has achieved a significant weight reduction compared to the original analog circuit, and is expected to achieve spaceborne applications.

Two suggestions:

       (1)The precision of microwave cavity temperature control is a key factor in achieving long-term stability indicators of hydrogen maser. The paper uses digital temperature control to achieve a temperature change of 0.013 degrees in the microwave cavity when the room temperature changes by 1.9 degrees, which basically meets the long-term stability requirements of hydrogen maser, but there is still potential for further improvement. Considering that the hydrogen maser operates in a vacuum environment in satellite application scenarios, there will be changes in heat transfer and dissipation characteristics. It is recommended to carry out optimization of temperature control design under vacuum in subsequent research.

      (2)CAT voltage and microwave cavity temperature changes are closely related. It is recommended to normalize Figure (C) and Figure (D) in Figure 16, provide their correlation, and conduct a specific analysis.

Author Response

Thank you very much for your attention and valuable comments on our paper. This manuscript has been revised carefully according to your comments. Your contents were reconsidered and confirmed in the revised manuscript.

Round 2

Reviewer 2 Report

Comments and Suggestions for Authors

The points I raised have been adequately addressed and I have no further comments.

Reviewer 3 Report

Comments and Suggestions for Authors

Dear authors,

Thank you for the revised version. I have no more comments. 

Comments on the Quality of English Language

I have no more comments.